# A Novel Device for Micro-Droplets Generation Based on the Stepwise Membrane Emulsification Principle

**DOI:** 10.3390/mi15091118

**Published:** 2024-08-31

**Authors:** Lei Lei, Sven Achenbach, Garth Wells, Hongbo Zhang, Wenjun Zhang

**Affiliations:** 1School of Integrated Circuit Science and Engineering, Beihang University, Beijing 100191, China; leileiearth@163.com; 2Department of Mechanical Engineering, University of Saskatchewan, Saskatoon, SK S7N 5A9, Canada; 3Department of Electrical and Computer Engineering, University of Saskatchewan, Saskatoon, SK S7N 5A9, Canada; s.achenbach@usask.ca; 4Synchrotron Laboratory for Micro and Nano Devices, Canadian Light Source Incorporated, Saskatoon, SK S7N 2V3, Canada; garth.wells@lightsource.ca; 5School of Mechanical and Power Engineering, East China University of Science and Technology, Shanghai 201620, China

**Keywords:** membrane emulsification process, microspheres generation, stepwise structure, photolithography, SU-8

## Abstract

This paper presents a novel design of the device to generate microspheres or micro-droplets based on the membrane emulsification principle. Specifically, the novelty of the device lies in a proposed two-layer or stepwise (by generalization) membrane structure. An important benefit of the stepwise membrane is that it can be fabricated with the low-cost material (SU-8) and using the conventional lithography technology along with a conventional image-based alignment technique. The experiment to examine the effectiveness of the proposed membrane was conducted, and the result shows that microspheres with the size of 2.3 μm and with the size uniformity of 0.8 μm can be achieved, which meets the requirements for most applications in industries. It is noted that the traditional membrane emulsification method can only produce microspheres of around 20 μm. The main contribution of this paper is thus the new design principle of membranes (i.e., stepwise structure), which can be made by the cost-effective fabrication technique, for high performance of droplets production.

## 1. Introduction

There are many applications for microspheres, e.g., cosmetics, electronics, pharmaceuticals, food processing, biomedicine, and drug delivery [1,2,3,4,5,6,7,8,9,10,11,12]. Microfluidic devices based on the emulsification principle with a membrane and/or micro-channel structure are increasingly of interest due to their high throughputs and high ability to control internal morphology. The requirements for any device that generates microspheres include (1) the size of the microspheres (the smaller, the better), (2) the uniformity of the microspheres in terms of size (the more uniform, the better), (3) the flexibility—i.e., the capability of making microspheres of different batch sizes with one device (the larger the range of the batch sizes, the better), and (4) the efficacy of the microspheres generation process (mass production or not). For comparison of various microsphere generation methods, refer to Lei et al. [13], Wu et al. [14], Arun K. Chattopadhyay and K.L. Mittal [15], and Solans et al. [16]. The general idea for all these devices that generate microspheres is to let two fluids interact with each other so that droplets may form in one of the two fluids [13], see Figure 1. The fluid out of which droplets are formed is called the dispersed phase, and the fluid that interacts with the dispersed phase is called the continuous phase (Figure 1).

Nakashima and Shimizu in the 1980s introduced a unique approach for generating emulsion by using a membrane of numerous uniform micro-pores [17]. The dispersed phase was forced by a low pressure to flow through the pores of the microporous membrane to interact directly with the continuous phase [18]. Emulsified droplets were formed from the dispersed phase and detached at the end of the pores with a drop-by-drop mechanism, as described in Figure 1. By this technique, the size of the droplet is mainly determined by the size of the pores in the porous membrane. Practically, the size of the generated droplet is about 2 to 10 times larger than size of the pore in the membrane [19]. Therefore, to obtain microspheres of small sizes, the challenge is to construct a membrane with many pores of small sizes (preferably in nanometers). There is a dilemma with the structure of the membrane. On one hand, a small pore membrane usually leads to a thin membrane with conventional membrane materials, which is fragile, while on the other hand, a thick membrane with small pores means that the membrane contains the high-aspect-ratio geometric feature, which is a challenge for microfabrication.

To tackle the above dilemma, the conventional idea in the literature is to seek unconventional materials, which can be made into a thin yet ductile and strong membrane. In connection with this, materials such as Shirasu Porous Glass (SPG) [17], silicon wafer [20], ceramic [21], stainless steel [22], and polymer [23] have been proposed in the literature. Among them, SPG was the most promising. However, fabrication of an SPG membrane is complex with high cost [17]. Moreover, the chance of contamination of an SPG membrane is high due to tortuous pore structures. Indeed, the poor rigidity and stability of a polymer membrane are the major shortcomings for membrane emulsification. For ceramic membranes, it is difficult to achieve well-defined circular openings, and there is a high chance of delamination during cleaning [21]. For stainless steel membranes, though they have excellent structural rigidity, small size holes on the membrane are difficult to achieve. Numerous other materials, e.g., silicon [20], etc., have been reported as membrane materials, yet difficulties in fabrication and high costs are still a challenge.

In the study reported in this paper, we propose a completely different idea from the above-mentioned literature; rather than focusing on the special types of materials, we focused on the structure of the membrane. Specifically, we propose a membrane with a stepwise structure such that we expect that a conventional material such as SU-8 can be used while the aspect ratio remains adequate to the conventional micro-fabrication technology.

## 2. Materials and Methods

### 2.1. Design

Figure 2 illustrates the concept of the stepwise structure, where Figure 2a illustrates a one-layer structure of the membrane, while Figure 2b illustrates a stepwise structure of the membrane. The design as illustrated in Figure 2a has two problems. The first problem is the difficulty in fabrication of such a “deep well” feature with the current photolithography (even with the expensive deep x-ray lithography process) [24]. The second problem is the difficulty of driving the dispersed phase fluid to flow through the long path channel [22]. It may be clear that the design concept of Figure 2a is not suitable to our constraint, namely use of a conventional photolithography (UV light as light source) process, and therefore, the design concept goes to one as illustrated in Figure 2b.

From Figure 2b the hole in the membrane is divided into several holes in different layers which have different thicknesses; specifically, the hole size is reduced stepwise from the top layer to the bottom layer with the smallest hole size being in the bottom layer. As such, the aspect ratio, an important parameter challenge in the photolithography process, corresponds to the smallest hole in the bottom layer, i.e., t_1_/w_po_, and the design should be adequate to the conventional photolithography process at our facility.

There were two design constraints for the embodiment design of the stepwise membrane in our study: (1) the membrane should be subject to the 120 kPa pressure estimated according to [20], and (2) the smallest hole size should be around 1 micron to generate microspheres of around 1 micron. The dimensions of the stepwise membrane in Figure 2b were determined based on a trial-end-error process along with a finite element simulation to meet these constraints. Depending on the photolithography process used, an adequate aspect ratio is noted, but the smaller the aspect ratio, the less challenging the photolithography process. In our case, the aspect ratio should be less than 5, and we chose 1 only. It is noted that the smallest hole size is also restricted by the resolution of a particular photolithography process. A size of 1 micron is about the limit. Thus, in Figure 2b, the membrane has two layers: the upper layer with the thickness (t_2_) and with the square pore size (w_pi_) and the bottom layer with the thickness (t_1_) and the square pore size (w_po_). The dispersed fluid will enter through the inlet pores (w_pi_) in the upper layer and exit through the narrow outlet pores (w_po_) in the bottom layer. Further, the pores in the bottom layer connect with the continuous phase fluid, which meets the dispersed phase fluid at the holes in the membrane. In this design, the thick upper layer was expected to withstand the pressure, i.e., meeting the constraint (1), and the smallest pore size in the bottom layer was expected to generate small (around 1 micron) microspheres, i.e., meeting the constraint (2).

For the study presented in this paper, the geometry of the stepwise membrane was determined based on the simulation alongside the experience as follows (Figure 2b): w_po_ = t_1_ = 1 µm and w_pi_ = t_2_ = 20 µm. The overall thickness (t) of the membrane was 21 µm (Figure 2a). Figure 3 shows the three-dimensional view of the stepwise membrane. As a result of this design, the aspect ratio of the upper layer was 20:20 (1:1) and that of the bottom layer was 1:1. Therefore, the conventional photolithography process could readily fabricate the device.

### 2.2. Fabrication

In this study, the materials (SU-8 2000 series) were used as the photoresist. SU-8 2000 series are the improved formulations of SU-8 and were suitable for the structure with its thick film (from 0.5 to >200 μm) and high aspect ratio (>10:1) [25,26]. The details of the photolithography process for fabricating the stepwise membrane are presented in the next sections.

#### 2.2.1. Chromium Sputtering

The glass substrate had the thickness of 100 μm, and it was sputtered with a layer of chromium with the thickness of 120 nm. The chromium layer was non-transparent so that the pores and the alignment marks were visible through the transparent glass substrate and SU-8 layer.

#### 2.2.2. SU-8 Spin Coating (1 μm)

SU-8 2001 was dynamically dispensed on the chromium, 1500 revolutions per minute (rpm) for 3 s (s) and then 4000 rpm for 30 s. The membrane was then soft baked on a hotplate at 95 °C for 1 min.

#### 2.2.3. Mark Alignment

Two alignment crosses were used as markers to help the alignment of two layers of SU-8, as shown in Figure 4.

#### 2.2.4. Exposure

The membrane was exposed under the laser with 12,000 $mJ/{cm}^{2}$. The laser power was 100 mW and the wavelength was 355 nm (DWL 66+, Heidelberg Instruments, Heidelberg, Germany). The direct write (maskless) lithography was used.

#### 2.2.5. Post-Exposure Bake (PEB)

The membrane was baked on the same hotplate at 95 °C for 1 min.

#### 2.2.6. Development

The membrane was developed in propylene glycol methyl ether acetate (PGMEA) for 1 min, and then it was rinsed off by isopropyl alcohol (IPA) for 30 s. At last, the membrane was dried by nitrogen ($N_{2}$). The opaque region was removed. Pores with the length of 1 × 1 μm^2^ were obtained, as shown in Figure 5.

#### 2.2.7. Oxygen Descum Plasma Cleaning

The flow rates of oxygen and argon were 50 and 20 standard cubic centimeters per minute (SCCM), respectively. The power for plasma cleaning was 150 W, and the time was 5 min (Diener Electronic, Germany).

#### 2.2.8. Chromium Etching

Chromium in the pore region was etched off by etching in Cr-etch for 45 s (MicroChemicals GmbH, Germany) and then rinsed with distilled (DI) water. The membranes before and after chromium etching are shown in Figure 6.

#### 2.2.9. SU-8 Spin Coating (20 μm)

SU-8 2025 was dynamically spin coated onto the membrane at 500 rpm for 1 s. Then the spin speed was manually accelerated to 4000 rpm over 10 s and kept for 30 s. The membrane was then soft baked on the same hotplate at 65 °C for 1 min. Then, the temperature was ramped to 95 °C and kept for 5 min.

#### 2.2.10. Mark Alignment

Two alignment crosses were used as marks to help the alignment of two layers of SU-8. As shown in Figure 7, the alignment mark of the first layer was covered by the alignment mark of the second layer perfectly.

#### 2.2.11. Exposure

The membrane was exposed under the laser with 12,000 $mJ/{cm}^{2}$. The laser power was 100 mW and the wavelength was 355 nm (DWL 66+, Heidelberg Instruments, Germany). The direct write (maskless) lithography was used.

#### 2.2.12. Post-Exposure Bake (PEB)

The membrane was baked on the same hotplate at 65 °C for 1 min. Then the temperature was ramped to 95 °C and kept for 5 min at 95 °C.

#### 2.2.13. Development

The membrane was developed in PGMEA for 4 min and then rinsed with IPA for 30 s. At last, the membrane was dried by $N_{2}$. The opaque region was removed. Pores with $20\times 20\;{\mu m}^{2}$ were obtained. The optical images of the top view and bottom view of the pores with $20\times 20\;{\mu m}^{2}$ are presented in Figure 8.

#### 2.2.14. Glued to an Acrylonitrile Butadiene Styrene (ABS) Tube

Epoxy 907 AB glues (Miller-Stephenson Chemical Incorporate, Danbury, CT, USA) were mixed for 3 min and then left to sit for 30 min. Next, the mixed glues were put around the edges of ABS tubing with 1 cm diameter (Taobao Company, Hangzhou, China), and then the tubing was placed on the top of samples, as shown in Figure 9. Twelve hours were needed, to wait for the best adhesive property.

#### 2.2.15. Substrate and Chromium Removing

On the first day, the membrane was immersed in 5% hydrofluoric (HF) acid for 8 h. On the second day, the membrane was etched by 5% HF acid for another 5 h to fully remove the glass substrate. Then the chromium layer was etched off by etching in Cr-etch for 3 to 4 min and then rinsed with distilled (DI) water and dried by $N_{2}$. Three pieces of membrane were obtained for the microsphere generation, as shown in Figure 10a. Two membranes failed during the gluing step because the glue covered these two membranes. The optical image of one piece of membrane is shown in Figure 10b. The diameter of the membrane emulsification device was 0.5 cm.

To sum up, the fabrication process of the porous membrane is shown in Figure 11.

### 2.3. Experiment

Paraffin oil (Sigma-Aldrich Corporation, St. Louis, MO, USA) with 2% Span 80 surfactant (Sigma-Aldrich Corporation, USA) was used as the continuous phase fluid and distilled (DI) water as the dispersed phase fluid in this experiment. Figure 12 shows the experimental set-up. A magnetic stirrer (Corning PC 210, Corning Incorporated, USA) was used to provide the shear force. A syringe pump (Harvard Apparatus, Holliston, MA, USA) controlling the flow rate of the dispersed phase was connected to the ABS tube by a syringe (6 mL, Covidien Limited, Dublin, Ireland) and tubing (1.34 mm PTFE, Adtech Polymer Engineering Limited, Stroud, UK). The syringe pump was calibrated before the experiment. The inverted microscope (Olympus IX70, Olympus Corporation, Tokyo, Japan) was used to observe the microsphere generation process, and a computer with image processing software was used for post-processing. The diameter of one single microsphere was measured multiple times to eliminate the possible error during the measuring process. The experiment was performed at room temperature (20 °C ± 2 °C) at the Intelligent Systems Laboratory at the University of Saskatchewan.

The minimum pressure ($P_{min}$) that ensures the dispersed phase can be pressed through the porous membrane is given by
(1)$$P_{min}=\frac{4\gamma cos\theta }{d_{p}}$$
where γ is the interfacial tension between the continuous and dispersed phases; θ is the contact angle between the dispersed phase and the membrane surface in the continuous phase, as shown in Figure 1; $d_{p}$ is the pore diameter. In this work, $\gamma =3.65\times 10^{-3}N\cdot m^{-1}$ [27] and the contact angle θ between the dispersed phase and the membrane surface in the continuous phase (Figure 1) was assumed to be zero because the wall was set to be a non-wetting boundary. Therefore, at the minimum pressure $P_{min}=\frac{4\times 3.65\times 10^{-3}}{1\times 10^{-6}}=1.46\;kPa$, the corresponding flow rate of the dispersed phase is about $5\;\mu L\cdot h^{-1}$. In this work, the flow rate of the dispersed phase was set to be ten times larger than the minimum flow rate, namely $50\;\mu L\cdot h^{-1}$. The agitation speeds of the magnetic stirrer were 60, 80, and 100 rpm. An amount of 1 mL continuous fluid with microspheres generated was carefully extracted from the beaker to a clean glass petri dish for observation. The image processing software (“Analyze Particles” Function, ImageJ, Original version, National Institutes of Health, Stapleton, NY, USA) was used to improve the resolution of microscopic images and measure the diameter of microspheres on the petri dish [28].

## 3. Results and Discussion

From Figure 13, there were lots of white dots in the continuous phase, which were microspheres generated and suspended in the continuous phase. Figure 14a,b, and c are the optical images of the microspheres generated by the SU-8 membrane under the agitation speeds of 60, 80, and 100 rpm, respectively. Figure 15 presents the average diameters of the microspheres along with the standard deviations for the different agitation speeds. It can be seen from Figure 15 that the diameters of the microspheres were 11.0 ± 3.5 μm, 7.4 ± 3.3 μm, and 2.3 ± 0.8 μm at agitation speeds of 60 rpm, 80 rpm, and 100 rpm, respectively. Thus, it can be concluded that the higher the agitation speed, the smaller the diameter of the microspheres. This phenomenon can be explained by the fact that higher agitation speed results in higher shear stress.

To test the strength of the designed SU-8 porous membrane in this study, the flow rate of the dispersed phase was increased gradually from 50 μL · h ^−1^ by steps of 100 μL · h^−1^. The first membrane failure happened when the flow rate of the dispersed phase was 4500 μL · h ^−1^ and the agitation speed was 100 rpm, as shown in Figure 16 (white arrow). It is noted that this flow rate was comparable to other membranes found in the literature. For example, Dragosavac et al. [29] produced microspheres using a nickel membrane with the flow rate of the dispersed phase at 3000 μL · h^−1^; the minimum diameter of the microspheres was 100 μm. Van der Graaf et al. [30] produced microspheres using a silicon membrane with the flow rate of the dispersed phase at $1.23\;\mu L\cdot h^{-1}$; the minimum diameter of the microspheres was 35 μm. Piacentini et al. [31] produced microspheres using a nickel membrane with the flow rate of the dispersed phase at $6000\;\mu L\cdot h^{-1}$; the minimum diameter of the microspheres was 40 μm. Vladisavljevic and Williams [32] produced microspheres using a stainless steel membrane with the flow rate of the dispersed phase at $1.2\times 10^{7}\;\mu L\cdot h^{-1}$; the minimum diameter of the microspheres was 100 μm. In the previous work of our group, Song [20] produced microspheres using a silicon membrane with the flow rate of the dispersed phase at $1.4\times 10^{6}\;\mu L\cdot h^{-1}$; the minimum diameter of the microspheres was 1.6 μm. In that work, the dispersed phase fluid was found to be difficult to push through the holes, because the height (t) of the holes with respect to the diameter (w) of the holes was too large, see Figure 2a. To make the membrane work, a high pressure was needed. However, it was found that the membrane was fragile and easily fractured. Addressing that problem was one of the motivations for the idea of the stepwise membrane presented in this paper. With the design in Figure 2b, the two flow paths (Path 1: t_1_ with respect to w_po_; Path 2: t_2_ with respect to w_pi_) are short, especially Path 1.

Thus, compared with the other membranes in the literature, as previously mentioned, the small diameter of the microspheres with good size uniformity (i.e., 2.3±0.8 μm) was achieved (the flow rate of the dispersed phase was 50 μL $\cdot h^{-1}$; the agitation speed was 100 rpm) with the developed SU-8 membrane. It is noted that microgel variation could range down to 10~15% in the literature [33], when the diameters of the microgels are larger than 100 μm. The diameters of the microspheres in our work were about 3 μm, which means very small differences cause larger variation. Thus, error rather than variation was used to evaluate the size uniformity of the microspheres. It is noted that the flow rate of the dispersed phase in this experiment can certainly be increased, as the maximal flow rate of the dispersed phase for the developed SU-8 membrane was 4500 μL$\cdot h^{-1}$ under the agitation speed of 100 rpm (notice: the breakage of the membrane was due to the pressure on the membrane, which is further a function of the flow rate of the dispersed phase and the agitation speed). With a far lower flow rate of the dispersed phase and the agitation speed of 100 rpm, the agitation speed can be increased, which implies further improved performance of the microsphere generation process with the SU-8 membrane in terms of small size and high uniformity of microspheres.

## 4. Conclusions and Future Work

This paper presents a study on the design, fabrication, and testing of a new emulsification membrane. The novelty of this membrane was the stepwise structure, which ensured that the size of the pores was small while the strength of the membrane was high. The membrane was made of the material SU-8, taken as the photoresist in the photolithography microfabrication process. The membrane had two layers. The first layer was the thickness of 1 µm, and the diameter of the pores in this layer was 1 µm. The second layer had the thickness of 20 µm, and the diameter of pores in this layer was 20 µm. The experiment was conducted to show the generation of microspheres with this novel membrane, which showed improved performance of the microsphere generation process with the SU-8 membrane in terms of the small size and the high uniformity of the microspheres. The experimental result showed that the new membrane is very promising in terms of the size of the microspheres as well as the size uniformity; particularly, the diameter and the uniformity of the microspheres were 11.0±3.5 μm, 7.4±3.3 μm, and 2.3±0.8 μm, under the agitation speeds of 60, 80, and 100 rpm, respectively. The stepwise membrane, as developed in this work, is cost effective, because the material of SU-8 photoresist is inexpensive and easy to manufacture, specifically fabricated with the conventional photolithography process, in comparison with the material of SPG along with its fabrication process [34].

To the best of our knowledge, this is the first study in the literature to use the SU-8 photoresist to make a membrane for microsphere generation. There are other factors that may influence the microsphere generation, such as surface roughness, pore alignment, interfacial tension between the continuous phase and dispersed phase, and so forth. These factors are worthy of future work. There is also interest in optimizing the design of the stepwise membrane.

## Figures and Tables

**Figure 1 micromachines-15-01118-f001:**
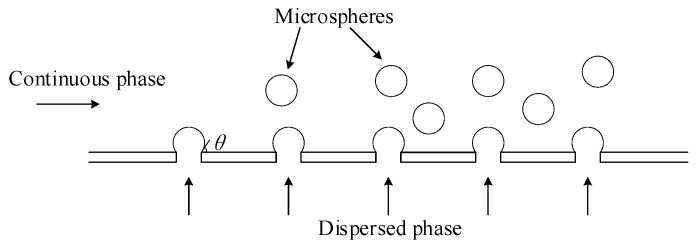
A side view of a membrane emulsification device.

**Figure 2 micromachines-15-01118-f002:**
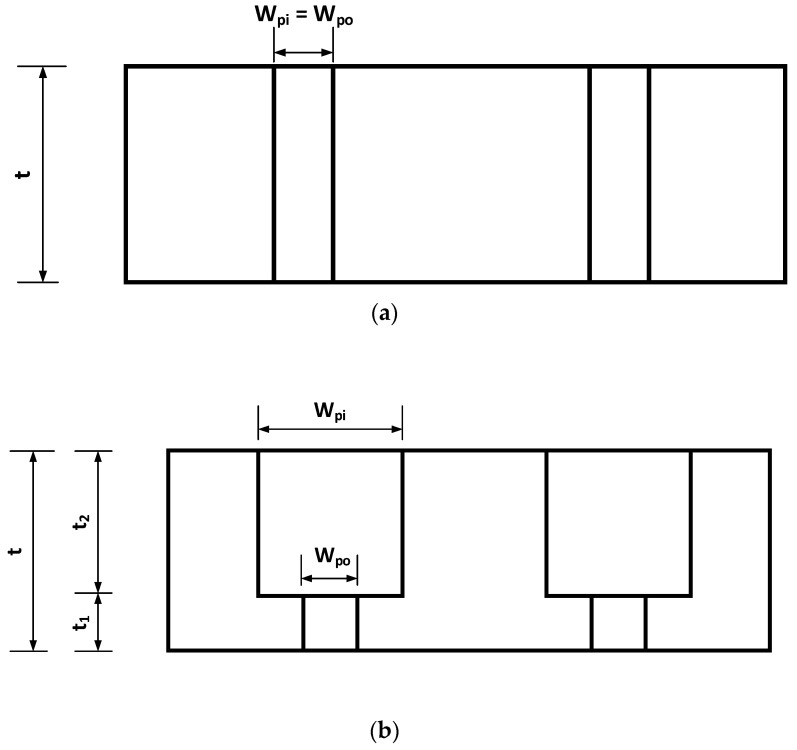
The membrane for microspheres generation. (**a**) The single-layer structure of the pores on the membrane emulsification device; (**b**) the stepwise structure of the pores (not to scale).

**Figure 3 micromachines-15-01118-f003:**
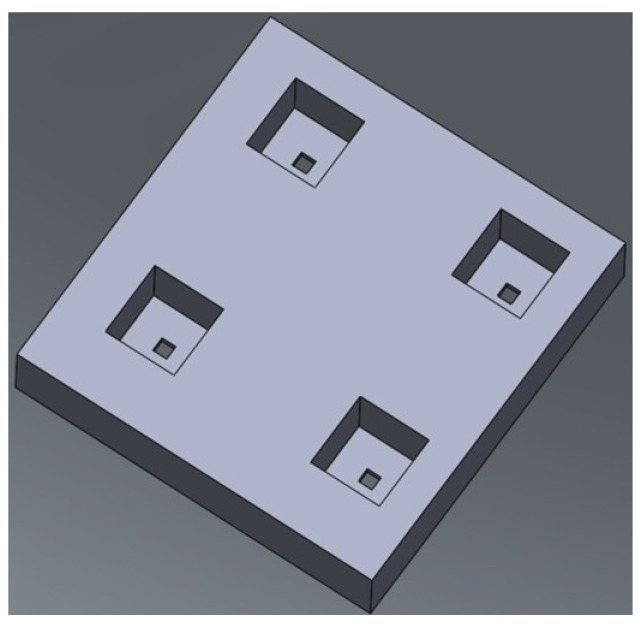
The stepwise pore structure of the new membrane emulsification device (not to scale).

**Figure 4 micromachines-15-01118-f004:**
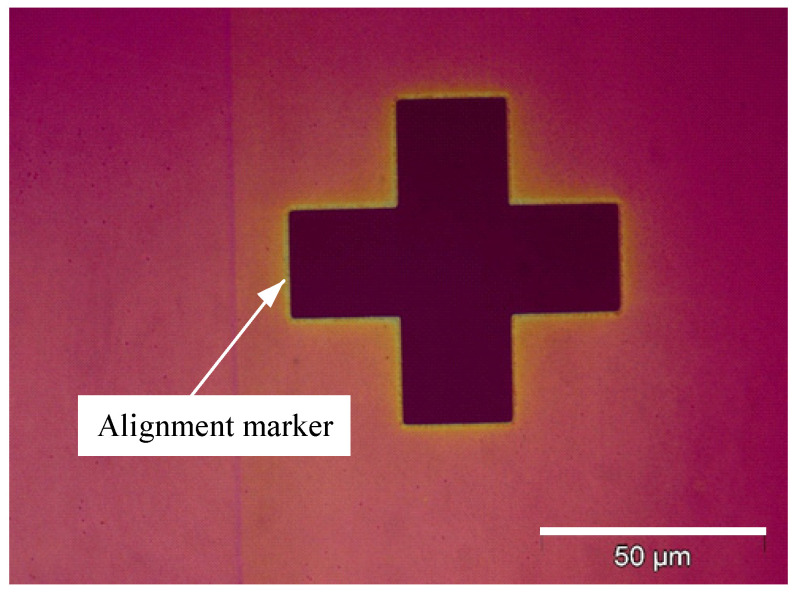
The optical image of one alignment on the first layer.

**Figure 5 micromachines-15-01118-f005:**
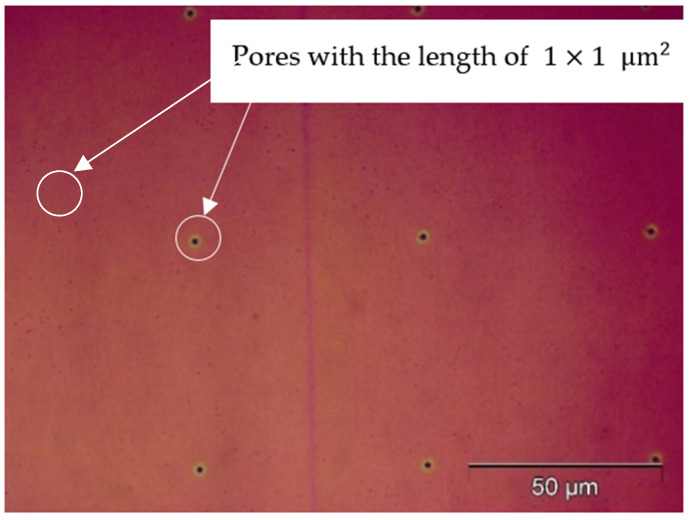
The optical image of pores with 1 × 1 μm^2^.

**Figure 6 micromachines-15-01118-f006:**
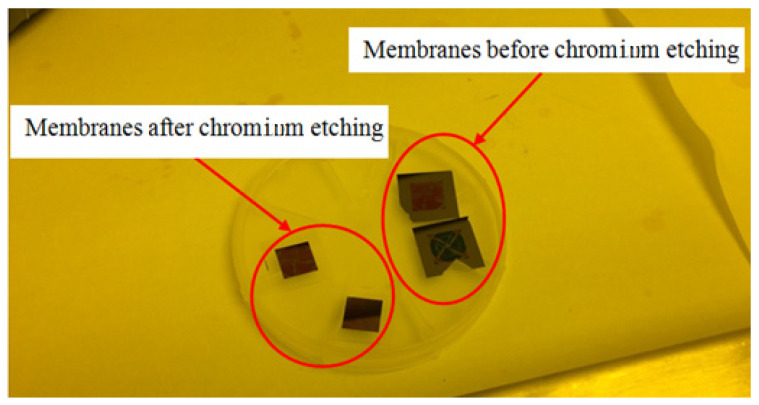
The membranes before and after chromium etching.

**Figure 7 micromachines-15-01118-f007:**
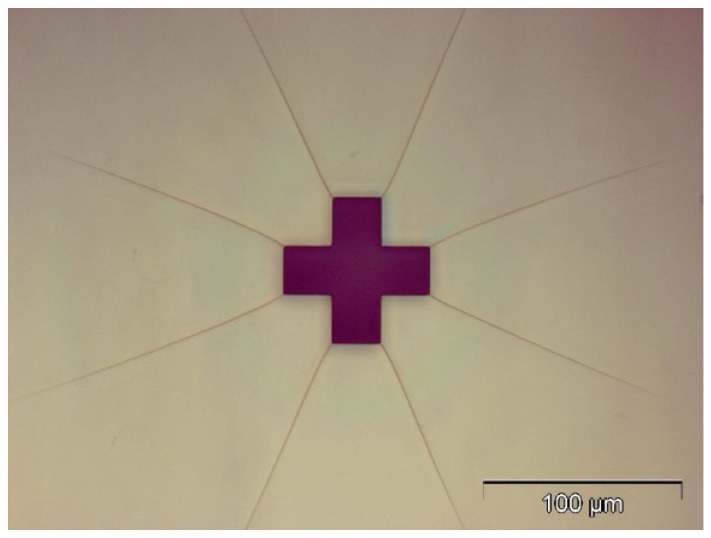
The optical image of one alignment on the second layer.

**Figure 8 micromachines-15-01118-f008:**
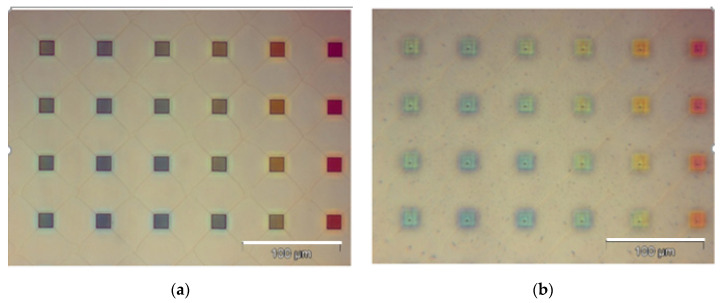
The optical images of (**a**) the top view and (**b**) bottom view of the pores with 20 × 20 μm^2^.

**Figure 9 micromachines-15-01118-f009:**
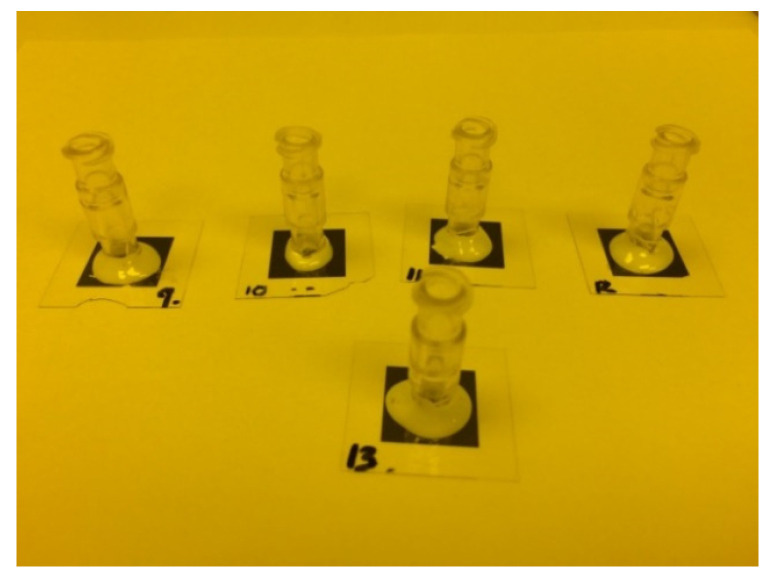
The membranes glued to ABS tubes. The diameter of the tubes is 1 cm.

**Figure 10 micromachines-15-01118-f010:**
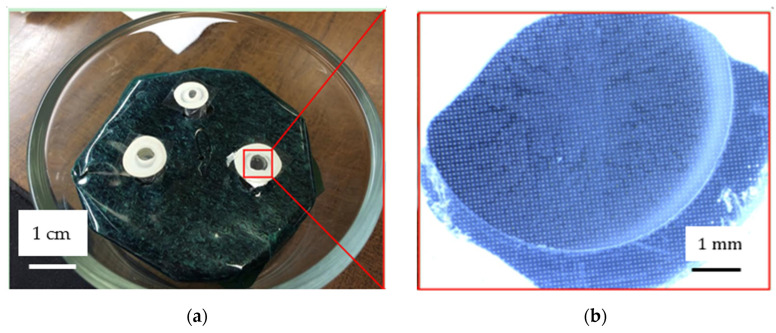
The fabricated stepwise membranes for microsphere generation. (**a**) Three membranes; (**b**) the magnified optical image of the porous membrane.

**Figure 11 micromachines-15-01118-f011:**
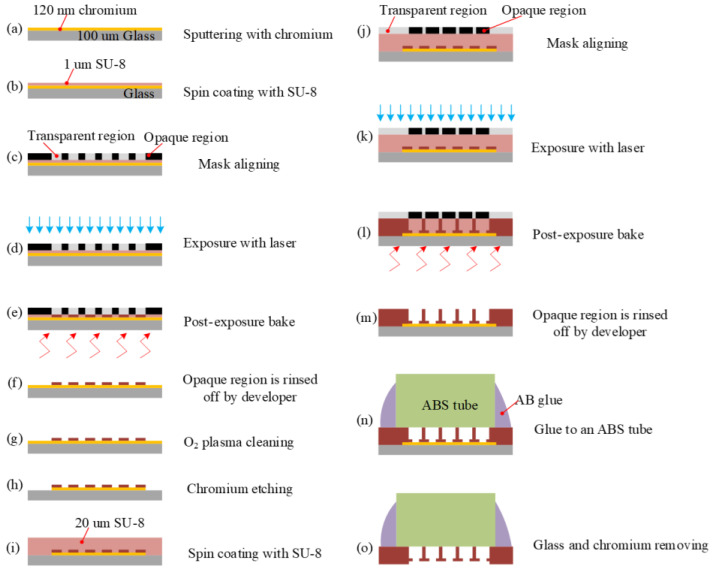
The schematic illustration of the fabrication procedure of the membrane with multilayer structure (not drawn to scale). (**a**) Sputtering with chrome; (**b**) Spin coating with SU-8; (**c**) Mask aligning; (**d**) Exposure with laser; (**e**) Post exposure bake; (**f**) Opaque region is rinsed off by developer; (**g**) O_2_ plasma cleaning; (**h**) Chrome etching; (**i**) Spin coating with SU-8; (**j**) Mask aligning; (**k**) Exposure with laser; (**l**) Post exposure bake; (**m**) Opaque region is rinsed off by developer; (**n**) Glue to an ABS tube; (**o**) Glass and chrome removing.

**Figure 12 micromachines-15-01118-f012:**
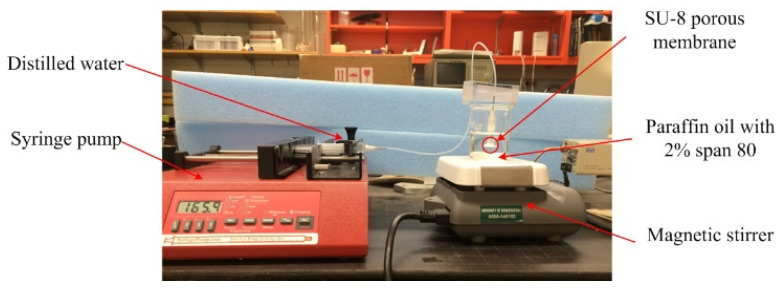
Experimental set-up for the membrane emulsification process.

**Figure 13 micromachines-15-01118-f013:**
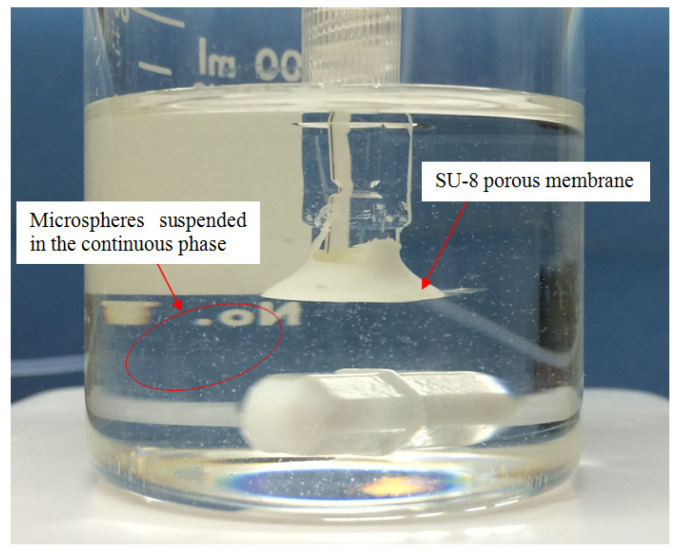
White dots are microspheres generated using the SU-8 membrane. Agitation speed: 100 rpm. Distance between the agitator and the membrane: 1 cm.

**Figure 14 micromachines-15-01118-f014:**
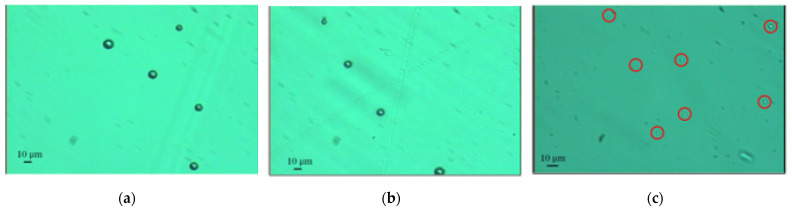
The optical images of the microspheres generated by the SU-8 membrane with the agitation speeds of 60 rpm (**a**), 80 rpm (**b**), and 100 rpm (**c**). Microspheres are highlighted in red circles in (**c**).

**Figure 15 micromachines-15-01118-f015:**
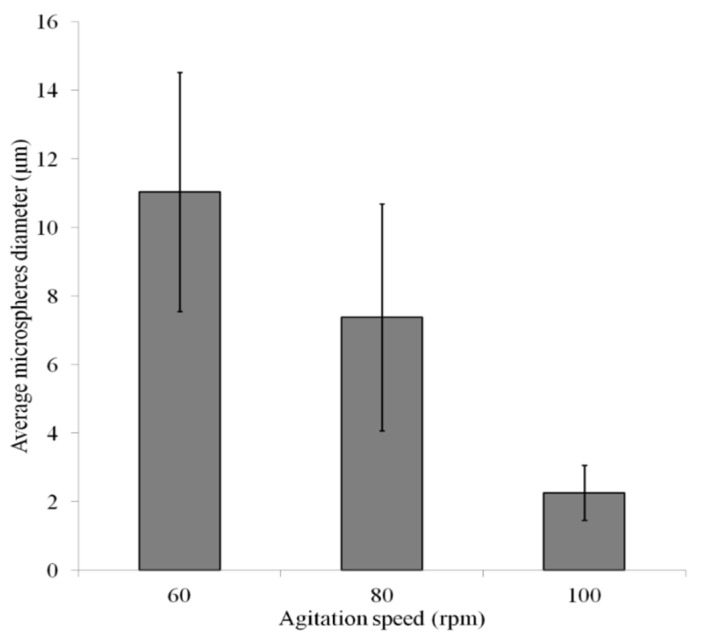
Average microsphere diameters with standard deviations for different agitation speeds.

**Figure 16 micromachines-15-01118-f016:**
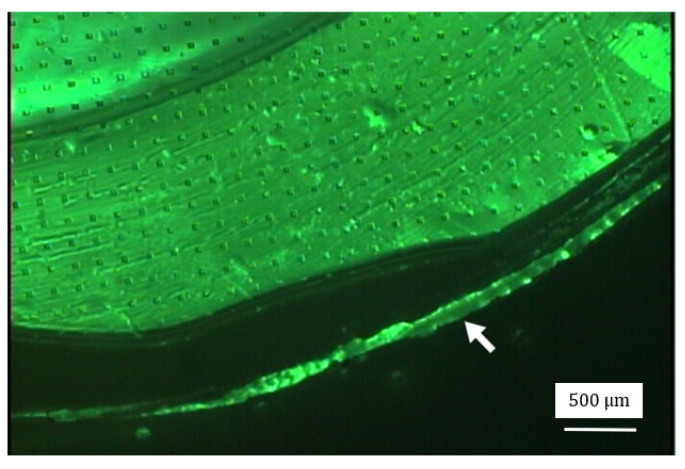
An optical image of the failure of the SU-8 membrane.

## Data Availability

The raw data supporting the conclusions of this article will be made available by the authors on request.

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
