# Peer review of "A Novel Device for Micro-Droplets Generation Based on the Stepwise Membrane Emulsification Principle"

_micromachines, 2024, doi:10.3390/mi15091118_

Round 1

Reviewer 1 Report

Comments and Suggestions for Authors

The manuscript requires minor revision

Review of manuscript No: micromachines-3130204 titled “A Novel Step-Wise Membrane Emulsification Device for Micro-Droplets Generation

Comments to Authors

This is a nice paper and worth be published in Micromachines, but after minor revision.

Fallowing should be revised:

1.      Please add a list of all abbreviations used in your work

2.      In the Results and Discussion Part pages 9 and 10, there is probably the mistake in the photos Figure 14, the diameter of the droplets  on the photo 14 c is the biggest it should be the smallest

3.      Could you please give information what was the ratio of the dispersed phase into the continuous phase?

4.      Please give the information about the polydispersity of the obtained droplets according to the agitation speed?

Author Response

Comment 1: Please add a list of all abbreviations used in your work

Response 1.1:Thanks for the comment. We have added “Nomenclature” section in our manuscript.

Comment 2: In the Results and Discussion Part pages 9 and 10, there is probably the mistake in the photos Figure 14, the diameter of the droplets on the photo 14 c is the biggest it should be the smallest

Response 1.2:Thank you for the meticulous revie. Yes, the diameter of droplets in Fig.14 (c) is the smallest. They are very small in the figure, and actually they look like dots in the figure. Therefore, we highlight these droplets in red circles. It can be seen that with the same scale, the diameter of droplets is smaller and smaller in Fig.14 (a), (b) and (c).

Comment 3: Could you please give information what was the ratio of the dispersed phase into the continuous phase?

Response 1.3:Sorry, we use a stirrer to form the continuous phase. Therefore, we cannot evaluate the flow rate of continuous phase directly, but we estimate this ratio based on the flow amount of these two phases, which is 1×10-4.

Comment 4: Please give the information about the polydispersity of the obtained droplets according to the agitation speed?

Response 1.4:Fig. 15 gives the standard deviation of droplets under different agitation speeds, which are , , and . These data are listed in “Conclusions” section.

Reviewer 2 Report

Comments and Suggestions for Authors

Please rephrase the abstract as it has some relevant content regarding the novelty of the paper.

Can you emphasize the percentage of benefit or advancement with the microsphere produced using this technique? In the conclusion section   

Can you put some high-magnification images of the micrographs?

How is the wall surface roughness?   

Do you observe clogging in the micro gaps? Is there any observation on the speed of the continuous and dispersed flow? 

Author Response

Comment 1: Please rephrase the abstract as it has some relevant content regarding the novelty of the paper.

Response 2.1:Thank you for the advice. Yes. I have noticed some vague descriptions in the Abstract of the original version of the manuscript. I revised the Abstract in the revised manuscript.

Comment 2: Can you emphasize the percentage of benefit or advancement with the microsphere produced using this technique? In the conclusion section   

Response 2.2:Thank you for your great suggestion. The impact of our work in terms of droplets generation is described further in the last section “Conclusion”. For the reviewer’s convenience, we quoted from the revised manuscript as follows:

Quoted:

The developed membrane is cost effective because the material of SU-8 photoresist is inexpensive and easy to manufacture with the conventional micro-fabrication technique. To the best of author’s knowledge, it is the first time to use SU-8 photoresist to make the membrane for microsphere generation.

Comment 3: Can you put some high-magnification images of the micrographs?

Response 2.3:Apology. All the figures in our manuscript are original ones captured by the microscope we used. The microscope may be updated to improve its resolution, which is beyond our research at this point of time. We could make statements in the future work section for this, if the reviewer can provide the latest information about the microscope.

Comment 4: How is the wall surface roughness?   

Response 2.4:This is really a great question. Unfortunately, we have not measured the roughness of the membrane. This is because of the scope of the work, as defined per se, is the feasibility of this stepwise membrane. The optimization of the design of the membrane in terms of the surface quality was considered out of the scope of our work at that time. However, we discussed the need of this work in the final section of the revised manuscript. For the reviewer’s convenience, we quoted the description from the revised manuscript here.

Quoted:

However, there are other factors may influence the microsphere generation, such as the surface roughness, the pore alignment, the interfacial tension of the continuous phase and dispersed phase, etc. These factors may need to be considered in the future work for further optimization.

Comment 5: Do you observe clogging in the micro gaps? Is there any observation on the speed of the continuous and dispersed flow? 

Response 2.5:This is a great question. We used paraffin oil and distilled water as continuous phase and dispersed phase, respectively. These two fluids were not found to clog the pores was well as channels. However, other polymeric compounds, e.g., PLGA, PLA, etc., may clog the pores as well as channels. Again, this is an interesting future work in this area.

Reviewer 3 Report

Comments and Suggestions for Authors

General

It is not clear why Sections 2, 3 and 4 are separate. Usually all experimental methodology is described in one section.

Abstract

In general, the abstract should be rewritten to emphasize further what was done, rather than why it is important.

Line 24 “an experiment was conducted”- Readers of a scientific journal article should already know that the authors conducted experiments! This sentence should instead describe which experiment was conducted.

Line 29 “cost-effective fabrication technique”- The technique described by the authors, producing a membrane from expensive photoresist with two separate photolithography steps plus metal deposition and etching, appears to be extremely labor- and resource-intensive. It does not seem like it can accurately be described as “cost-effective”.

Introduction

The beginning of the introduction needs to be clearer. Is emulsification the only method of producing microspheres? If so, this needs to be stated- if not, the advantages of emulsification need to be discussed. The authors should also be clearer about different forms of emulsification- as well as the membrane-based droplet formation discussed here, droplets can be formed using microfluidic devices including those designed for high throughput (see the review by Wu et al, Appl Phys Rev 8, 031304 (2021), doi:10.1063/5.0049897), as well as more conventional methods based on stirring (See JL Salager in Surfactants in Solution, CRC Press, 2020) or spontaneous emulsification (see Solans et al, Curr Opin Colloid Interf. Sci. 22, 88-93 (2016), doi: 10.1016/j.cocis.2016.03.002). The authors need to compare the advantages and disadvantages of these methods.

The authors should also cite the review by Charcosset et al (J Chem Technol Biotechnol 79:209-218 (2004), doi:10.1002/jctb.969) as an introduction to the principles of membrane emulsification for readers unfamiliar with the field.

Lines 46-47: Usually in an emulsion the two components are referred to as dispersed and continuous phase, rather than flow.

Line 84: Can SU-8 be described as a conventional material? While it is widely used in research, it is expensive (in excess of US$1000 per litre) and the use of spin coating means that large amounts of SU-8 are required to coat a wafer. To the best of the reviewer's knowledge there is no commercial product using SU-8 as a material (as opposed to as a resist for e.g. semiconductor production). While the authors have demonstrated that this membrane can be fabricated in a specialized lab, they do not discuss whether it could be feasibly produced for commercial/industrial applications, compared to membrane materials such as SPG which can be produced using a much more scalable chemical process.

Lines 88-91: If the study is conventionally organized (Introduction, Materials and Methods, Results and Discussion, Conclusion) this does not need to be described in the paper.

Design Section 2

Line 95: “It may be clear”- either it is clear, or it isn’t! The authors should explain better what figure 2 shows. In addition, they should explain that their design constraint (1) in line 98 entails a minimum thickness for the membrane. It may be better to redraw Fig. 2 giving hypothetical examples of a membrane with sufficient thickness and pore size (but an aspect ratio presenting fabrication challenges), sufficient thickness and aspect ratio (but unacceptably large pores) and sufficient aspect ratio and pore size (but insufficient thickness)

Line 106: How did the authors determine that a 20 µm membrane thickness was enough to withstand the pressure? Were simulations or experiments done? If so, these need to be detailed. Alternatively, if this thickness value was obtained from literature, it needs to be cited.

Line 119-20: It would be more correct to say that both layers have an aspect ratio of 1:1.

Fabrication Section 3

 In general, this section is too long and detailed. The use of a mask aligner (or maskless laser writer) is a standard technique, so for instance the images of alignment crosses do not need to be shown. In addition, the separate steps do not need their own subsection. However, some important details are also missing.

Line 131: Where was the 100 µm glass substrate obtained? Was it standard wafer scale? If not, what were the dimensions?

Line 132: Surely the chromium layer was transparent?

Lines 138, 149, 171, 185: Why were two different types of hotplate used? If one was covered and the other was not, this should be stated. Otherwise, the specific brand of hotplate is unnecessary detail.

Line 146: Exposure for photolithography is usually expressed in mJ/cm2 or equivalent units not mW- a power figure without an exposure time or area does not give sufficient information on the level of exposure. In addition, it should be made clearer that direct write (maskless) lithography was used.

Line 159: Which parameters were used for the descum (as well as time, power, pressure and oxygen flow are important)? What type of system was used?

Lines 153, 190: “Opaque region was removed”- Does this refer to the standard process of developing with PGMEA until no white opaque streaks appear when rinsing with IPA? If so, this is unnecessary detail. If not, it needs to be rephrased to be more understandable.

Line 200: What are the dimensions and source of the ABS tubing used?

Figure 9: Given the perspective in the figure, the scale bar here cannot be correct for all the devices shown. For this type of image, the usual practice is to only show one device, and to place a coin next to it for scale (either with the image taken in such a way that readers can identify what coin it is, or the coin identified and its diameter stated in the caption).

Figure 10b: The scale bar here seems like it must be incorrect given the dimensions stated in the text.

Line 209: “Three pieces of membranes were obtained”- This appears to be a statement of the yield of the process. The fact that 5 devices are shown in Fig. 9 suggests that the process failed at different points, and only 3 working devices were obtained at the end. The authors need to state how many devices would be obtained from 1 run of the production process assuming all steps took place perfectly, and how many failed at each step.

Lines 224-229: This should be in the acknowledgments section not the Methods section.

Figure 11: Some parts of the text in this image appear to be blurred/pixelated. It should be replaced by a higher quality version.

Section 4

Line 242: ImageJ has functions allowing for the automated measurement of feature size. Were these functions used to measure microsphere diameter? If the microsphere diameter was measured manually instead, why was this done?

Line 244: The location of the experiment is irrelevant.

Line 263: ImageJ is not commercial software! It is open-source government funded software. The authors need to cite the relevant publication according to the specific version of ImageJ used- see here: https://imagej.net/contribute/citing

Section 5

Line 273: How many microspheres were measured for each experiment? Were replicate experiments conducted, or does each result correspond to a single experiment? The authors should conduct replicate experiments if this has not already been done. In addition, replicates should be performed with different membranes at the same agitation speed, and with different agitation speeds and the same membrane, to rule out any difference in microsphere size due to variations in the membrane.

Figure 15: If the red circles in Figure 15c are aids to allow readers to see the microspheres, this needs to be stated in the caption. In general, this is a poor quality image- could it be improved?

Line 296: Is this actually comparable? In particular, note that the authors in ref. 20 were able to produce smaller microspheres with a flow rate more than 300 times the maximum shown in this paper.

Line 313: Is the size uniformity “good”? The coefficient of variation here is 35%. Microfluidics-based methods for microgel production give a significantly lower CV. The CV obtained here is comparable to those obtained for microgels produced by conventional batch-emulsion methods without the need for a membrane.

see Muir et al., ACS Biomater Sci Eng.  (2021) 7(9): 4269–4281, doi: 10.1021/acsbiomaterials.0c01612

The review by Charcosset et al. mentioned earlier states that even 20 years ago, CV values of 10% were obtained by membrane emulsification with SPG membranes.

Conclusion

Lines 341-342: As explained earlier, it is not correct to describe SU-8 photoresist as inexpensive or easy to manufacture.

Lines 346-348: This final paragraph appears to be taken from a completely different paper! It does not seem to bear any relation to the rest of the manuscript, and it is not clear what it is doing here.

References

It would be more helpful to readers if all references had DOIs where applicable. Where the authors cite a PhD thesis, this should be made clear in the reference. 

Comments on the Quality of English Language

Intro:

Line 26 “which meets most of applications”- should be “which meets requirements for most applications”

 Line 49: pours should be pores

Line 55 should say “size of the pore”

Line 60-61 should be “contains high aspect ratio geometric features, which are challenging for microfabrication

Line 67 should be “…made into a thin yet ductile…”

The use of “to” in lines 69-77 (e.g. “to SPG”) is not correct- “For” is perhaps better.

Line 82 should be “the special materials used by other researchers”

Line 189: “Had a PGMEA bath” is not scientific English. “Was developed in PGMEA” would be more correct. Similarly, “bathing” (line 162) and “taking a Cr-etch bath”- it is more correct to say “Chromium was removed by etching in Cr-etch for [time]”

Figure 10 caption: Surely the image is magnified rather than amplified?

Line 276: Better to phrase as “higher agitation speed results in higher shear stress”

Line 296: “comparative” should be “comparable”

Line 348: “smoot” should be “smooth”

Author Response

General

Comment 1: It is not clear why Sections 2, 3 and 4 are separate. Usually, all experimental methodology is described in one section.

Response 3.1:Thank you for the suggestion. In the revised manuscript, we designed a section called “Materials and Methods”, Section 2. This section has three sub-sections: Design, Fabrication, Experiment. Section 3 is “Results with Discussions”. Section 4 is “Conclusion and Future Work”.

Abstract

Comment 2:In general, the abstract should be rewritten to emphasize further what was done, rather than why it is important.

Response 3.2:Thank you for this great suggestion. We completely rewrote the abstract in the revised manuscript.

Comment 3: Line 24 “an experiment was conducted”- Readers of a scientific journal article should already know that the authors conducted experiments! This sentence should instead describe which experiment was conducted.

Response 3.3:Thank you for this great suggestion. We revised this sentence to “The experiment to examine the effectiveness of the proposed membrane was conducted, and the result shows that microspheres with the size of 2.3 μm and with the size uniformity of 0.8 μm can be achieved, which meets requirements for most applications in industries.”

Comment 4: Line 29 “cost-effective fabrication technique”- The technique described by the authors, producing a membrane from expensive photoresist with two separate photolithography steps plus metal deposition and etching, appears to be extremely labor- and resource-intensive. It does not seem like it can accurately be described as “cost-effective”.

Response 3.4:Thank you for this very valuable comment, as indeed the cost includes both material and fabrication process. In literature, to create a hole around 1 micron or smaller, an option of the material is called SPG. The material cost of SU-8 is about 10% of the cost of SPG. SU-8 material can be fabricated with the conventional photolithography process, as we did, and this process is nowadays widely available. However, the fabrication of SPG membrane was very complicated at the time our research was carried out. In fact, to our best knowledge, there is only one company in Japan that can fabricate the SPG membrane.

Introduction

Comment 5: The beginning of the introduction needs to be clearer. Is emulsification the only method of producing microspheres? If so, this needs to be stated- if not, the advantages of emulsification need to be discussed. The authors should also be clearer about different forms of emulsification- as well as the membrane-based droplet formation discussed here, droplets can be formed using microfluidic devices including those designed for high throughput (see the review by Wu et al, Appl Phys Rev 8, 031304 (2021), doi:10.1063/5.0049897), as well as more conventional methods based on stirring (See JL Salager in Surfactants in Solution, CRC Press, 2020) or spontaneous emulsification (see Solans et al, Curr Opin Colloid Interf. Sci. 22, 88-93 (2016), doi: 10.1016/j.cocis.2016.03.002). The authors need to compare the advantages and disadvantages of these methods.

Response 3.5:Again, a great comment to improve our paper. We significantly revised the introduction section, including the comment on the literatures as suggested and they appear in the References section.

Comment 6: The authors should also cite the review by Charcosset et al (J Chem Technol Biotechnol 79:209-218 (2004), doi:10.1002/jctb.969) as an introduction to the principles of membrane emulsification for readers unfamiliar with the field.

Response 3.6:Thank you for the suggestion. We revised the introduction and included the valuable literature in the reference section.

Comment 7: Lines 46-47: Usually in an emulsion the two components are referred to as dispersed and continuous phase, rather than flow.

Response 3.7:Thanks for the comment. We have revised the expression.

Comment 8: Line 84: Can SU-8 be described as a conventional material? While it is widely used in research, it is expensive (in excess of US$1000 per litre) and the use of spin coating means that large amounts of SU-8 are required to coat a wafer. To the best of the reviewer's knowledge there is no commercial product using SU-8 as a material (as opposed to as a resist for e.g. semiconductor production). While the authors have demonstrated that this membrane can be fabricated in a specialized lab, they do not discuss whether it could be feasibly produced for commercial/industrial applications, compared to membrane materials such as SPG which can be produced using a much more scalable chemical process.

Response 3.8:It is true that SU-8 was used in a laboratory setting rather than an industrial scale at the time our research was carried out. The scope of our research at that time was to respond to the question of building a membrane with small holes (around 1 micron or less). We found the SPG material along with its fabrication process. However, we found that SPG is too high cost, as discussed above, and besides, its fabrication process is unique and not readily available in our university (University of Saskatchewan, Canada). We were looking for the material with moderate cost and readily fabricated with popular micro-fabrication techniques, photolithography in this case. It is noted that the photo-lithography process can make an arrange of micro-membranes, as opposed to the spin coating process. Other materials, which may be cheaper than SU-8 and can be made by photolithography, were considered out of the scope of our work. However, from a commercialization point of view, it is worthy of effort to examine other materials. 

Comment 9: Lines 88-91: If the study is conventionally organized (Introduction, Materials and Methods, Results and Discussion, Conclusion) this does not need to be described in the paper.

Response 3.9: Thanks for the comment. We have removed this part. In the revised manuscript, the organization is as the reviewer indicated.

Design Section 2

Comment 10: Line 95: “It may be clear”- either it is clear, or it isn’t! The authors should explain better what figure 2 shows. In addition, they should explain that their design constraint (1) in line 98 entails a minimum thickness for the membrane. It may be better to redraw Fig. 2 giving hypothetical examples of a membrane with sufficient thickness and pore size (but an aspect ratio presenting fabrication challenges), sufficient thickness and aspect ratio (but unacceptably large pores) and sufficient aspect ratio and pore size (but insufficient thickness)

Response 3.10:Thank you for the comment, which helps to improve the paper a lot. Following the reviewer’s suggestion, we re-drew Figure 2 in the revised manuscript. According to the systematic design methodology, Figure 2 shows the result of the so-called embodiment design, at which the geometry of the design concept (i.e., stepwise membrane) is determined. The dimension of the stepwise device was determined based on the simulation (initially trial-and-error) to make sure that the stress in the membrane is not over the tensile stress of the material. Depending on the photolithography process used, an adequate aspect ratio is noted, but the smaller the aspect ratio, the less challenging the photolithography process. In our case, the aspect ratio should be less than 5, and we chose 1 only. It is noted that the smallest hole size is also restricted by the resolution of a particular photolithography process. The size of 1 micron is about the limit.       

Comment 11: Line 106: How did the authors determine that a 20 µm membrane thickness was enough to withstand the pressure? Were simulations or experiments done? If so, these need to be detailed. Alternatively, if this thickness value was obtained from literature, it needs to be cited.

Response 3.11:We chose the overall thickness (t) of the membrane to be 21 um based on rough estimation, specifically based on a schematic model of the membrane plate subject to a uniform pressure (120 kPa) on top of it. The strength test was performed to show that this overall thickness of the membrane has sufficient strength to be subject to the pressure on the membrane. Due to the scope of our study, i.e., demonstrating the feasibility of the concept of stepwise membrane to generate microspheres of around 1 micron, we did not study how thickness may affect the quality of the process of microspheres generation. However, this can be an interesting future work, mentioned in the revised manuscript.    

Comment 12: Line 119-20: It would be more correct to say that both layers have an aspect ratio of 1:1.

Response 3.12:Thanks for the comment. We have revised the expression. However, the optimal determination of these parameters may be possible, which we consider is a future work, mentioned in the revised manuscript.

Fabrication Section 3

In general, this section is too long and detailed. The use of a mask aligner (or maskless laser writer) is a standard technique, so for instance the images of alignment crosses do not need to be shown. In addition, the separate steps do not need their own subsection. However, some important details are also missing.

Comment 13: Line 131: Where was the 100 µm glass substrate obtained? Was it standard wafer scale? If not, what were the dimensions?

Response 3.13:It is standard wafer.

Comment 14: Line 132: Surely the chromium layer was transparent?

Response 3.14:The chromium layer was non-transparent.

Comment 15: Lines 138, 149, 171, 185: Why were two different types of hotplate used? If one was covered and the other was not, this should be stated. Otherwise, the specific brand of hotplate is unnecessary detail.

Response 3.15:Thanks for the comment. We have removed the brand of hotplate.

Comment 16: Line 146: Exposure for photolithography is usually expressed in mJ/cm2 or equivalent units not mW- a power figure without an exposure time or area does not give sufficient information on the level of exposure. In addition, it should be made clearer that direct write (maskless) lithography was used.

Response 3.16:Thanks for the comment. We have added more information about the exposure.

Comment 17: Line 159: Which parameters were used for the descum (as well as time, power, pressure and oxygen flow are important)? What type of system was used?

Response 3.17:Thanks for the comment. We have added more information about this step.

Comment 18: Lines 153, 190: “Opaque region was removed”- Does this refer to the standard process of developing with PGMEA until no white opaque streaks appear when rinsing with IPA? If so, this is unnecessary detail. If not, it needs to be rephrased to be more understandable.

Response 3.18:Yes, it is standard process. We only want to express the results of this step as well as make sure the step can be repeated by the reader.

Comment 19: Line 200: What are the dimensions and source of the ABS tubing used?

Response 3.19:The diameter of ABS tube is 1 cm, and we have added this information in the revised manuscript.

Comment 19: Figure 9: Given the perspective in the figure, the scale bar here cannot be correct for all the devices shown. For this type of image, the usual practice is to only show one device, and to place a coin next to it for scale (either with the image taken in such a way that readers can identify what coin it is, or the coin identified, and its diameter stated in the caption).

Response 3.20:Thanks for the comment. We agree that the suggested way to preset this information better. However, since this study was performed several years ago, the bench for the lab, etc. has been disbanded.

Comment 21: Figure 10b: The scale bar here seems like it must be incorrect given the dimensions stated in the text.

Response 3.21:Thanks for the comment. It should be mm, not cm. We have corrected this error in the revised manuscript.

Comment 22: Line 209: “Three pieces of membranes were obtained”- This appears to be a statement of the yield of the process. The fact that 5 devices are shown in Fig. 9 suggests that the process failed at different points, and only 3 working devices were obtained at the end. The authors need to state how many devices would be obtained from 1 run of the production process assuming all steps took place perfectly, and how many failed at each step.

Response 3.22:Thanks for the comment. We have illustrated the reason why 2 out of 5 membranes were failed in the revised manuscript.

Comment 23: Lines 224-229: This should be in the acknowledgments section not the Methods section.

Response 3.23:Thanks for the comment. We have put this part in the acknowledgement section in the revised manuscript. 

Comment 24: Figure 11: Some parts of the text in this image appear to be blurred/pixelated. It should be replaced by a higher quality version.

Response 3.24:Sorry, all the figures shown in our manuscript are original versions captured by the microscope. They are the highest magnification images we could get. Since this study was performed several years ago, the bench for the lab, etc. has been disbanded, and use of the latest image capturing facility is difficult.

Section 4

Comment 25: Line 242: ImageJ has functions allowing for the automated measurement of feature size. Were these functions used to measure microsphere diameter? If the microsphere diameter was measured manually instead, why was this done?

Response 3.25:We measured the diameter of microspheres automatically using functions in ImageJ. But the software may be recently updated, which unfortunately beyond our research.

Comment 26: Line 244: The location of the experiment is irrelevant.

Response 3.26:We agreed with the reviewer, but other reviewers may want us to give as complete information as possible. In our campus, there are several places our experiment can be carried out.  

Comment 27: Line 263: ImageJ is not commercial software! It is open-source government funded software. The authors need to cite the relevant publication according to the specific version of ImageJ used- see here: https://imagej.net/contribute/citing

Response 3.27:Thanks for the comment. We have corrected this error in the revised manuscript.

Section 5

Comment 28: Line 273: How many microspheres were measured for each experiment? Were replicate experiments conducted, or does each result correspond to a single experiment? The authors should conduct replicate experiments if this has not already been done. In addition, replicates should be performed with different membranes at the same agitation speed, and with different agitation speeds and the same membrane, to rule out any difference in microsphere size due to variations in the membrane.

Response 3.28:The amount of 1 ml continuous fluid with microspheres generated was carefully extracted from the beaker to a clean glass petri dish for observation. Therefore, the number of microspheres generated was random. We have not used different membranes to replicate experiments, since the membrane was inspected carefully before the microsphere generation process, and the dimensions of the membranes were found quite close. Nevertheless, there may be an issue of robustness, that is, how sensitive is with the proposed approach, i.e., stepwise membrane based on membrane emulsification principle to generate microspheres?    

Comment 29: Figure 15: If the red circles in Figure 15c are aids to allow readers to see the microspheres, this needs to be stated in the caption. In general, this is a poor-quality image- could it be improved?

Response 3.29:The microspheres in Fig. 14(c) are very small in the figure, and indeed, they look like dots in the figure, so we highlight these droplets in red circles. We have added this illustration in the revised manuscript.

Comment 30: Line 296: Is this actually comparable? In particular, note that the authors in ref. 20 were able to produce smaller microspheres with a flow rate more than 300 times the maximum shown in this paper.

Response 3.30:In the revised manuscript, it is Ref. 24. In Ref. 24 (a previous work in our group), the dispersed phase fluid was found difficult to be pushed through the hole, because the height (t) of the hole with respect to the diameter (w) of the hole is too large, see Fig. 2a. To make the membrane work, a high pressure is needed. However, it was found that the membrane was fragile and easily fractured. Addressing this problem is one of the motivations of the idea of stepwise membrane presented in this paper. With the design in Fig. 2b, two flow paths (Path 1: t1 with respect to wpo; Path 2: t2 with respect to wpi) are short, especially with Path 1.               

Comment 31: Line 313: Is the size uniformity “good”? The coefficient of variation here is 35%. Microfluidics-based methods for microgel production give a significantly lower CV. The CV obtained here is comparable to those obtained for microgels produced by conventional batch-emulsion methods without the need for a membrane. see Muir et al., ACS Biomater Sci Eng.  (2021) 7(9): 4269–4281, doi: 10.1021/acsbiomaterials.0c01612. The review by Charcosset et al. mentioned earlier states that even 20 years ago, CV values of 10% were obtained by membrane emulsification with SPG membranes.

Response:3.31: The microgel variation is 10~15% in the mentioned reference, while the diameters of microgels are larger than 100 um. The diameter of microspheres in our work is around 1-3 um. It is noted that CV will tend be large for small microspheres, which is the principle, namely, the smaller the microsphere, the larger the CV. It is worth mentioning that CV represents the reliability of the production. This principle is in fact the general principle in manufacturing of small objects, e.g., chip. The mentioned paper is included in the reference section.

Conclusion

Comment 32: Lines 341-342: As explained earlier, it is not correct to describe SU-8 photoresist as inexpensive or easy to manufacture.

Response 3.32:We compare costs between SU-8 and SPG membrane, which both produce microspheres with smaller size than conventional membrane (silicon, metal, etc.). The cost of SU-8 is about 1/10 of SPG membrane, and the fabrication of SU-8 membrane is conventional photolithography technique. While the fabrication of SPG membrane is very complicated, thus only one company in Japan can fabricate this membrane, that is also the reason why SPG membrane is expensive. Thus, we say SU-8 membrane is cost-effective compared with SPG membrane. In the revised manuscript, we highlight the comparison between SU-8 and SPG membrane.

Quoted:

“The developed membrane is cost effective because the material of SU-8 photoresist is inexpensive and easy to manufacture, specifically with the conventional micro-fabrication technique, in comparison with SPG.”

Comment 33: Lines 346-348: This final paragraph appears to be taken from a completely different paper! It does not seem to bear any relation to the rest of the manuscript, and it is not clear what it is doing here.

 Response 3.33:Thanks for the comments. We have removed this part.

References

Comment 34: It would be more helpful to readers if all references had DOIs where applicable. Where the authors cite a PhD thesis, this should be made clear in the reference. 

 Response 3.34:Thanks for the comment, but we refer to the template given by the journal, which does not include DOIs in the references. PhD thesis is indicated in the entry in the reference section.

Comments on the Quality of English Language

Intro:

Comment 35: Line 26 “which meets most of applications”- should be “which meets requirements for most applications”

 Response 3.35:Thanks for the comment. We have revised the expression.

Comment 36:  Line 49: pours should be pores

 Response 3.36:Thanks for the comment. We have revised the expression.

Comment 37: Line 55 should say “size of the pore”

 Response 3.37:Thanks for the comment. We have revised the expression.

Comment 38: Line 60-61 should be “contains high aspect ratio geometric features, which are challenging for microfabrication

 Response 3.38:Thanks for the comment. We have revised the expression.

Comment 39: Line 67 should be “…made into a thin yet ductile…”

 Response 3.39:Thanks for the comment. We have revised the expression.

Comment 40: The use of “to” in lines 69-77 (e.g. “to SPG”) is not correct- “For” is perhaps better.

 Response 3.40:Thanks for the comment. We have revised the expression.

Comment 41: Line 82 should be “the special materials used by other researchers”

 Response 3.41:Thanks for the comment. We have revised the expression.

Comment 42: Line 189: “Had a PGMEA bath” is not scientific English. “Was developed in PGMEA” would be more correct. Similarly, “bathing” (line 162) and “taking a Cr-etch bath”- it is more correct to say “Chromium was removed by etching in Cr-etch for [time]”

 Response 3.42:Thanks for the comment. We have revised the expression.

Comment 43: Figure 10 caption: Surely the image is magnified rather than amplified?

 Response 3.43:Thanks for the comment. We have revised the expression.

Comment 44: Line 276: Better to phrase as “higher agitation speed results in higher shear stress”

 Response 3.44:Thanks for the comment. We have revised the expression.

Comment 45: Line 296: “comparative” should be “comparable”

 Response 3.45:Thanks for the comment. We have revised the expression.

Comment 46: Line 348: “smoot” should be “smooth”

 Response 3.46:Thanks for the comment. We have removed this part.

Round 2

Reviewer 3 Report

Comments and Suggestions for Authors

The authors are thanked for their response to the review and for the changes they have made. Please note that I have not replied to all the comments, only those where I consider that further revisions are necessary.

Response 3.4: Thank you for the useful insight. If SU-8 is indeed 10% of the cost of SPG, then it is fair to describe this method for membrane fabrication as low cost. However, this needs to be made clearer in the article, ideally with a suitable reference.

Response 3.8: The added paragraph (Lines 91-103 of the revised manuscript, discussing the shortcomings of current membrane materials) is a welcome addition to the paper. However, the referencing here must be improved. Reference 22 is cited both referring to SPG membrane fabrication (line 96) and delamination of ceramic membranes (line 100), when the reference does not seem to mention either of these. Similarly, Reference 21 is cited in reference to silicon wafer-based membranes (lines 93 and 102) when this paper is in fact about glass membranes.

It is possible that something went wrong with the referencing software, resulting in citations in the text that point to the wrong references. If so, this needs to be fixed.

Response 3.10: This response clarifies the novelty of the authors’ method. However, it would be helpful to point out more clearly in the text that the hypothetical membrane shown in Figure 2a would present fabrication challenges due to its high aspect ratio.

Response 3.11: Which finite element simulation package was used (line 122)?

Response 3.13: The authors claim that the glass substrate was standard wafer dimensions (presumably 100mm diameter round?). However, the membranes shown in Figure 6 appear to be square, and not the shape of a standard wafer. Where does this difference in shape come from? Were multiple membranes fabricated on each substrate?

Response 3.16: What was the wavelength of the laser?

Response 3.20: While I accept that it may no longer be possible to reproduce this image, the scale bar is simply incorrect and should not be in there. If a new image cannot be taken, the authors should remove the scale bar and state the diameter of the tubes in the caption to give the reader a sense of scale.

Response 3.22: While this information is useful, the authors need to clarify/confirm (if this is the case) that if the fabrication had proceeded perfectly with no failures, 5 membranes would have been produced. If this is not the case, they need to explain how many membranes would have been produced if there were no failures, and what stage the remaining failures occurred at.

Response 3.24: Figure 11 is an illustration, it was not produced using a microscope. Therefore, this excuse makes no sense. The authors presumably still have access to whatever graphics software was used to draw this illustration, and could either redraw it or simply export it again at a higher DPI.

Response 3.25: The authors need to state which function of Image J (e.g. Analyze Particles) was used to measure microsphere diameter.

Response 3.27: While the authors have corrected the error, they should still cite the relevant publication for the ImageJ version they used.

Response 3.30: This needs to be made clearer in the manuscript. The authors should state that this was previous work in their lab, and describe the problems experienced with the membrane easily fracturing.

Further Changes:

Line 151: the word “high” needs to be added before “aspect ratio”.

Reference 12 is incomplete- the volume number is incorrect (it should be 11 not 23) and a page/article number (5300) is needed.

Reference 18 is incomplete- the journal title is incorrect, and volume/issue/page numbers are needed. The correct citation is Bulletin of the Ceramic Society of Japan, 1986, 21, (5), 408-412

Reference 22 is incomplete- a page/article number (101328) is needed as well as the journal, year and volume.

Reference 27 and 28 are incomplete- they need page/article numbers (425302 and 055001 respectively).

Comments on the Quality of English Language

Line 92: "In this connection" should be "in connection with this"

Line 100: "Getting" should be removed

Line 117: "wop" needs to be "wpo" as in the nomenclature section

Line 190: "were" should be "are"

Author Response

Comment 1: Comments and Suggestions for Authors

The authors are thanked for their response to the review and for the changes they have made. Please note that I have not replied to all the comments, only those where I consider that further revisions are necessary.

Thank you for the useful insight. If SU-8 is indeed 10% of the cost of SPG, then it is fair to describe this method for membrane fabrication as low cost. However, this needs to be made clearer in the article, ideally with a suitable reference.

Response 1: Thanks for your suggestion. We have cited this reference: [34] Mugabi, J., Jeong, JH. Review of the technological advances for the preparation of colloidal dispersions at high production throughput using microporous membrane systems. Colloid Polym Sci 302, 463–485 (2024). https://doi.org/10.1007/s00396-023-05217-8.

Comment 2: The added paragraph (Lines 91-103 of the revised manuscript, discussing the shortcomings of current membrane materials) is a welcome addition to the paper. However, the referencing here must be improved. Reference 22 is cited both referring to SPG membrane fabrication (line 96) and delamination of ceramic membranes (line 100), when the reference does not seem to mention either of these. Similarly, Reference 21 is cited in reference to silicon wafer-based membranes (lines 93 and 102) when this paper is in fact about glass membranes.

It is possible that something went wrong with the referencing software, resulting in citations in the text that point to the wrong references. If so, this needs to be fixed.

Response 2: Line 96 has been corrected to Ref. [18]. Ref. [22] is about ceramic membrane, so line 100 is correct. Line 93 and 102 have been corrected to Ref. [21].

Comment 3: This response clarifies the novelty of the authors’ method. However, it would be helpful to point out more clearly in the text that the hypothetical membrane shown in Figure 2a would present fabrication challenges due to its high aspect ratio.

Response 3:

Thanks for the comment. We explain this challenge in Section 2.1. For the reviewer’s convenience, we quoted from the revised manuscript as follows:

Quoted:

“The design as illustrated in Figure 2a has two problems. The first problem is the difficulty in fabrication of such a “deep well” feature with the current photolithography (even with the expensive deep x-ray lithography process) [24]. The second problem is the difficulty of driving the dispersed phase fluid to flow through the long path channel [22]. It may be clear that the design concept of Figure 2a is not suitable to our constraint, namely use of a conventional photolithography (UV light as light source) process, and therefore, the design concept goes to one as illustrated in Figure 2b.”

Comment 4: Which finite element simulation package was used (line 122)?

Response 4: We use FEM module available in the Solidworks software.

Comment 5: The authors claim that the glass substrate was standard wafer dimensions (presumably 100mm diameter round?). However, the membranes shown in Figure 6 appear to be square, and not the shape of a standard wafer. Where does this difference in shape come from? Were multiple membranes fabricated on each substrate?

Response 5: Yes, multiple membranes are fabricated on each substrate. We do lithography process on one substrate, and then cut the substrate into several membranes.

Comment 6: What was the wavelength of the laser?

Response 6: The membrane was exposed under the laser with the laser power of 100 mW and the wavelength of 355 nm (DWL 66+, Heidelberg Instruments, Germany). We added this information to the manuscript. For the reviewer’s convenience, we quoted from the revised manuscript as follows:

Quoted: The membrane was exposed under the laser with 12000  and the wavelength is 355 nm (DWL 66+, Heidelberg Instruments, Germany). The direct write (maskless) lithography was used.

Comment 7: While I accept that it may no longer be possible to reproduce this image, the scale bar is simply incorrect and should not be in there. If a new image cannot be taken, the authors should remove the scale bar and state the diameter of the tubes in the caption to give the reader a sense of scale.

Response 7: We have removed the scale bar, and state the diameter of the tubes in the caption for Fig. 9.

Comment 8: While this information is useful, the authors need to clarify/confirm (if this is the case) that if the fabrication had proceeded perfectly with no failures, 5 membranes would have been produced. If this is not the case, they need to explain how many membranes would have been produced if there were no failures, and what stage the remaining failures occurred at.

Response 8: Yes, we produced 5 membranes when the fabrication proceeded perfectly.

Comment 9: Figure 11 is an illustration; it was not produced using a microscope. Therefore, this excuse makes no sense. The authors presumably still have access to whatever graphics software was used to draw this illustration, and could either redraw it or simply export it again at a higher DPI.

Response 9: We have updated Fig. 11 with higher DPI.

Comment 10: The authors need to state which function of Image J (e.g. Analyze Particles) was used to measure microsphere diameter.

Response 10: We have stated the function (Analyze Particles) we used in this work.

Comment 11: While the authors have corrected the error, they should still cite the relevant publication for the ImageJ version they used.

Response 11: We have cited the following reference: [29] Schneider, C., Rasband, W. & Eliceiri, K. NIH Image to ImageJ: 25 years of image analysis. Nat Methods 9, 671–675 (2012). https://doi.org/10.1038/nmeth.2089.

Comment 12: This needs to be made clearer in the manuscript. The authors should state that this was previous work in their lab, and describe the problems experienced with the membrane easily fracturing.

Response 12: We have stated this situation in the manuscript. For the reviewer’s convenience, we quoted from the revised manuscript as follows:

Quoted:

“In the previous work of our group, Song [21] produced microspheres using a silicon membrane with the flow rate of the dispersed phase of ; the minimum diameter of microspheres was . In this work, the dispersed phase fluid was found difficult to be pushed through the hole, because the height (t) of the hole with respect to the diameter (w) of the hole is too large, see Figure 2a. To make the membrane work, a high pressure is needed. However, it was found that the membrane was fragile and easily fractured. Addressing this problem is one of the motivations of the idea of stepwise membrane presented in this paper. With the design in Figure 2b, two flow paths (Path 1: t1 with respect to wpo; Path 2: t2 with respect to wpi) are short, especially with Path 1.” 

Further Changes:

Comment 13: Line 151: the word “high” needs to be added before “aspect ratio”.

Response 13: We have revised here into: quoted: “In this study, the materials (SU-8 2000 series) were used as the photoresist. SU-8 2000 series were the improved formulations of SU-8, which were suitable for the structure with a thick film () and a high aspect ratio () [26, 27]. The details of the photolithography process for fabricating the stepwise membrane are presented in the next sections.”

Comment 14: Reference 12 is incomplete- the volume number is incorrect (it should be 11 not 23) and a page/article number (5300) is needed.

Response 14: It is Volume 23, Issue 11. The article number has been added.

Comment 15: Reference 18 is incomplete- the journal title is incorrect, and volume/issue/page numbers are needed. The correct citation is Bulletin of the Ceramic Society of Japan, 1986, 21, (5), 408-412

Response 15: We have corrected the citation.

Comment 16: Reference 22 is incomplete- a page/article number (101328) is needed as well as the journal, year and volume.

Response 16: We have added article number. The journal, year, and volume have been added in the original version.

Comment 17: Reference 27 and 28 are incomplete- they need page/article numbers (425302 and 055001 respectively).

Response 17: We have added article number.

Comment 18: Comments on the Quality of English Language

Line 92: "In this connection" should be "in connection with this"

Response 18: We have revised this sentence.

Comment 19: Line 100: "Getting" should be removed

Response 19: We have revised this sentence.

Comment 20: Line 117: "wop" needs to be "wpo" as in the nomenclature section

Response 20: We have revised this sentence.

Comment 21: Line 190: "were" should be "are"

Response 21: We have revised this sentence.